# Impact of Experimental Parameters on Cell–Cell Force Spectroscopy Signature

**DOI:** 10.3390/s21041069

**Published:** 2021-02-04

**Authors:** Reinier Oropesa-Nuñez, Andrea Mescola, Massimo Vassalli, Claudio Canale

**Affiliations:** 1Department of Materials Science and Engineering, Uppsala University, Ångströmlaboratoriet, Box 35, SE-751 03 Uppsala, Sweden; reinier.oropesa@angstrom.uu.se; 2CNR-Nanoscience Institute-S3, Via Campi 213/A, 41125 Modena, Italy; andrea.mescola@nano.cnr.it; 3James Watt School of Engineering, University of Glasgow, Glasgow G128LT, UK; 4Department of Physics, University of Genoa, via Dodecaneso 33, 16146 Genoa, Italy; canale@fisica.unige.it

**Keywords:** cell-cell interaction, force spectroscopy, atomic force microscopy, cell mechanics, mechanobiology

## Abstract

Atomic force microscopy is an extremely versatile technique, featuring atomic-scale imaging resolution, and also offering the possibility to probe interaction forces down to few pN. Recently, this technique has been specialized to study the interaction between single living cells, one on the substrate, and a second being adhered on the cantilever. Cell–cell force spectroscopy offers a unique tool to investigate in fine detail intra-cellular interactions, and it holds great promise to elucidate elusive phenomena in physiology and pathology. Here we present a systematic study of the effect of the main measurement parameters on cell–cell curves, showing the importance of controlling the experimental conditions. Moreover, a simple theoretical interpretation is proposed, based on the number of contacts formed between the two interacting cells. The results show that single cell–cell force spectroscopy experiments carry a wealth of information that can be exploited to understand the inner dynamics of the interaction of living cells at the molecular level.

## 1. Introduction

Cells have several evolved mechanisms to sense and respond to mechanical stimuli in their environment. Mechanical forces transmitted through cell–matrix and cell–cell interactions play a pivotal role in the organization, growth, maturation, and function of living tissues [1,2,3,4,5]. Cell–cell interactions are not only crucial to maintaining tissue morphogenesis and homeostasis, but they also activate signaling pathways important for the regulation of different cellular processes including, cell survival, cell migration, and differentiation [6,7]. Alterations in the plasma membrane composition, and consequently, its nanomechanical properties and the nanoscale forces arising from the cell–cell interactions, can impair cellular mechanosensitivity [8,9] and eventually lead to the onset of several human pathologies. Although cell–cell connections are commonly represented as two phospholipid bilayers tethered by a few receptors, the compartments created possess properties that are distinct from and more complex than other parts of the plasma membrane. Indeed, cell mechanics is affected during the pathological mechanisms in breast cancer diseases by the alteration of the expression of cell membrane components [10]. The interaction of Aβ42 oligomers was also found to negatively influence the membrane’s biophysics of hippocampal neurons [11]. Therefore, the understanding of the underlying molecular pathways of cell–cell interactions is a crucial aspect for a better comprehension of human pathologies.

In this context, mechanobiology has emerged as an active field to quantify the mechanics of cell–cell and cell–matrix interactions integrating biophysical measurements and technique developments ranging from the molecular to cellular level. Among the biophysical techniques in this field, atomic force microscopy (AFM) is an exciting analytical tool to measure the binding mechanics of cell–cell molecules. The AFM was originally used to obtain surface topography. Moreover, it precisely measures the interaction force between the probe tip and the sample surface with pico-nano Newton resolution. Different AFM approaches have been successfully applied to study cellular systems. The use of large colloidal or flat probes in indentation experiments allows the determination of the mechanical properties of a cell, mediating the result on a large contact area [12]. At the same time, the use of a sharp tip as indenter allows detection of local changes of mechanical properties [13,14] as well as investigation of single-molecule unfolding events [15]. The stiffer cytoskeletal filaments network [16] can be characterized by AFM, as well as the softer nuclear compartment [17].

Functional probes in molecular recognition force microscopy mode allowed for detecting specific ligand–receptor interaction forces [18] on cells. Although the size and shape can change significantly, the probe is always made by a rigid material, generally silicon or silicon nitride, in some cases coated/functionalized with a single molecular species. 

A more advanced methodology for single cell force spectroscopy has been further established to quantify cell–substrate adhesion [19]. The idea is to bind a living cell to an AFM tipless cantilever, using it as an extraordinarily powerful, but at the same time complex, probe. The presence of many specific binding sites and different non-specific interaction sources makes the analysis of force spectroscopy curves acquired in this modality a challenging task. For this reason, the technique has often been applied on simple and well-controlled samples; material substrates [20], substrates coated/functionalized with a single molecular species [21], or multifunctional substrates of very well-known molecules [22]. 

The cellular probe can be used to extend the analysis beyond the cell–substrate interaction, and studying with great detail cell–cell adhesion. Cell–cell force spectroscopy (CCFS) is a technique in which a cell attached to the tipless AFM cantilever is brought in contact with another living cell, while the interaction force is collected [23]. The quantitative evaluation of cell–cell interaction offers a powerful tool for biomedical research and it paves the way for future diagnostic translation. In fact, this experimental procedure can be further extended to probe cell–tissue interaction. Nowadays, the challenging idea to consider the adhesion between a particular cell with cells derived from a tissue possibly involved in a pathological state, such as cancer, as a diagnostic marker of the pathology is not far to be realized. 

Besides its great potential, CCFS has been the focus of only a relatively low number of papers published in the field. The complexity of cell–cell curves requires a particular attention in the analysis. In this context, it is fundamental to control the experimental conditions, having a clear understanding of how particular acquisition parameters can influence the results. Clear and standard methodologies are still not defined. In this work we present a systematic approach to CCFS. Using Chinese Hamster Ovary (CHO) cells and testing cell-cell interaction varying different experimental settings, aiming to disclose how these settings are influencing the results. 

## 2. Materials and Methods

### 2.1. Cell Culture

CHO (CCL-61T; ATCC, Teddington, UK) cell lines were cultured on Petri dishes (Techno Plastic Products, Neuchâtel, Switzerland), coated with poly-D-lysine (PDL; Sigma-Aldrich, Milano, Italy), in Dulbecco’s modified Eagle’s medium (Gibco, Paisley, UK) containing 4.5% glutamine and glucose, 10% inactivated fetal bovine serum, 1.0% penicillin-streptomycin, and 1.0% nonessential amino acids (Gibco) at 37 °C in 5.0% CO_2_. The cells were split every 4–5 days before reaching a confluency rate of <80%.

### 2.2. Cantilever Functionalization and Cell Capture

Single beam silicon tipless cantilevers TL1-50 (NanoWorld, Neuchâtel, Switzerland), with a nominal spring constant of 0.03 N/m, were irradiated in an ultraviolet/ozone cleaner (ProCleaner; Bioforce Nanosciences, Ames, IA, USA) for 15 min before functionalization. The cantilevers were functionalized with concanavalin A (ConA, Sigma-Aldrich, Milano, Italy) as previously described [24,25].

For cell attachment, CHO cells (density of 3 × 10^3^ cells mL^−1^) were removed from the Petri dish via trypsinization. Briefly, the culture medium was removed from the Petri containing the confluent cells, and the cells were first washed with sterile phosphate-buffered saline (PBS) and, subsequently, incubated with 0.5% trypsin-EDTA 0.05% (Gibco, Thermofisher, Milano, Italy) for 2 min at 37 °C. The trypsinized cells were resuspended in 1 mL of PBS buffer and centrifuged for 5 min at 200× *g*. After centrifugation, the cells were resuspended in PBS and gently agitated. A few detached cells were injected into a standard sterile Petri dish where CHO cells were cultured at a density of 3 × 10^3^ cells mL^−1^. Before cell seeding, a small part of the coverslip was coated with agarose, by spreading a 20 µL drop of 0.15% *w*/*w* agarose solution (Sigma-Aldrich, Milano, Italy) until gelification. The lack of adhesion between the cell and the repulsive agarose surface significantly increased the efficiency of the cell capture procedure. A single cell was captured by pressing for 30 s the functionalized cantilever onto a cell lying on the agarose spot with a controlled force of 2.0 nN, and then by lifting the cantilever. The system was left for 10 min to get a stable cell-cantilever contact before the acquisition of force–distance (F–D) curves.

### 2.3. Cell–Cell Force Spectroscopy (CCFS)

CCFS experiments were carried out using a Nanowizard III system (Bruker, JPK Instruments, Berlin, Germany), coupled with an AxioObserver D1 (Zeiss, Oberkochen, Germany) inverted optical microscope. A CellHesion module (Bruker, JPK Instruments) was used to extend to 100 μm the vertical displacement range of the AFM. All experiments were carried out at 37 °C in PBS containing 2.0 mM CaCl_2_ and 2.0 mM MgCl_2_ and setting the force-curve length at 80 μm to achieve complete detachment of the cell probe from the target cell. For each set of experiments, only the measurement condition in consideration was changed while keeping the rest of the parameters unaltered. All the acquired F–D curves were processed with the JPK Data Processing software to correct for the bending of the cantilever and to remove the baseline offset and exported in txt format for further analysis.

Approach and retract speeds: the approach and retract speeds were also analyzed. For all the cases, the contact was kept for 45 s using the constant-height as the delay mode (see below). The setpoint force, hence the maximum force exerted between the two cells, was set at 1.0 nN. For the evaluation of the approach speed influence in the cell–cell interaction, the cantilever was lifted at a constant velocity of 10.0 μm/s. Different approach speeds (1, 2.5, 10, 25, and 50 μm/s) were investigated. To understand how the retract speed influences the measurements, the approach speed was set at a constant velocity of 10.0 μm/s. After the contact, the cell probe was lifted at 1, 2.5, 10, 25, and 50 μm/s. Over 10 F–D curves per speed were acquired for every experiment, for a total of 53 and 354 curves for approach and retract speeds, respectively.

Delay time: different extended pauses (1, 15, 30, 45, 60, and 120 s) were studied. The cell probe was lowered at a constant speed of 10.0 μm/s until the cell probe contacted the target cell and the preset force (setpoint force) of 1.0 nN was reached. The constant height was used as the delay mode. The cell probe was then retracted, lifting the cantilever at a constant velocity of 10.0 μm/s. A total of 211 curves were acquired.

Setpoint force: three setpoint forces (1, 10, and 30 nN) were studied. The cell probe was lowered at a constant speed of 10.0 μm/s until the cell probe contacted the target cell and the preset force was reached. The contact was kept for 45 s using the constant height as the delay mode. Then the cell probe was retracted, lifting the cantilever at a constant velocity of 10.0 μm/s to register the F–D curve. At least 33 F–D curves per each setpoint were acquired for a total of 100 F–D curves.

Delay mode: the contact between the two cells can be maintained in two different modalities, constant-force and constant-height. In constant-force, after reaching the setpoint force, the piezo actuator compensates for any change of the interaction force associated with the cell shape’s adaption and remodeling under an applied load. In constant height mode, the piezo actuator maintains a fixed position after reaching the setpoint force. In this second case, the cell position is fixed, but the interaction force can change due to the cell’s ability to remodel its shape. The cell probe was lowered and retracted at a constant speed of 10.0 μm/s. The cell probe was kept in contact with the target cell for 45 s after the preset force of 1.0 nN was reached. A total of 52 F–D curves for constant force mode and 41 F–D curves for the constant height mode were analyzed.

### 2.4. Data Analysis

Force spectroscopy curves were exported from the JPK software to text format, and further analyzed using a custom software developed using Python 3 and the scientific libraries offered by NumPy/SciPy [26]. The source code of the software is currently available through github [27]. The software is designed to analyze the retract segment of each force–distance curve, batch processing a folder based on a selected set of parameters and exporting the results in a comma-separated values (CSV) text file for further statistics (see below). Each curve is processed to identify the baseline (based on the part of the curve far from the sample) and the origin of the Z is placed where the retract curve first crosses the baseline value; all distances are calculated with respect to this point. The curve is thus segmented, using peaks in the first derivative to identify discontinuity points (based on a Savitzky–Golay filter [28]. The final detachment point (Z_det_, F_det_) is thus identified as the last discontinuity point. The detachment work W (see Figure 1) is calculated as the area under the curve from Z = 0 to the detachment point Z = Z_det_.

## 3. Results

Cell–cell interactions were investigated by AFM-based force spectroscopy (hereafter called cell–cell force spectroscopy, CCFS). Generally, these experiments consist of force versus displacement curves obtained when a cell adhered to the tipless cantilever is brought in contact with a second cell seeded in the culture dish, as represented in Figure 1. The measure starts when the tipless cantilever, functionalized with the CHO cell, is settled above the target CHO cell but far from the sample. Then it is moved towards the sample with a constant approach speed *v_a_* (red line in Figure 1) until the preset force value *F_s_* (setpoint) is reached. It is relevant to keep this force under a few nN, in order to restrict the interaction between the cells to the membrane and cortical region, without causing any damage to the cells. Cells are kept in contact for different preset times (contact time τ) while keeping constant either the applied force or the position (delay mode). Subsequently, the AFM cantilever is pushed away from the surface with a constant retract speed *v_r_*, independent of the approach one. The retract segment (green line in Figure 1) shows many steps, associated with the rupture of specific bonds between the cells, and the experiment ends when the cells are fully detached (observed in the curve when a final plateau at force ~0 nN is reached). Each curve is then segmented, and a set of relevant parameters is extracted (see Section 2.4), including the detachment (or adhesion) work *W* that was further used to primarily quantify the cell-cell interactions. All these experimental parameters impact the corresponding detachment curve, and they can disclose different aspects of the cell–cell interaction.

The influence of the retraction speed was analyzed by varying *v_r_* while keeping constant all other parameters: approach speed *v_a_* = 10.0 μm/s, setpoint force *F*_0_ = 1.0 nN, contact time τ = 45 s, delay mode: constant height. The results are reported in Figure 2a where the detachment work *W* is plotted as a function of the retract speed. The distribution of measured *W* clearly shows a trend towards higher average values, accompanied by larger distributions and the appearance of a tail for large values in the distribution (Figure 2a). While the retraction speed is expected to impact the adhesion work largely, another less obvious and often neglected parameter was also investigated: the approach speed. A second set of experiments where the approach speed *v_a_* has been varied is reported in panel b) of Figure 2. All the other parameters have been kept constant: retract speed *v_r_* = 10.0 μm/s, contact time τ = 45 s, delay mode: constant height. Interestingly, the trend in this case suggests a decrease of the detachment work with increasing approach speed (Figure 2b) and no broadening of the distribution is apparent for this experimental configuration.

Other parameters that can influence cell–cell adhesion are the delay time τ and the setpoint force *F*_0_ (see Figure 2c,d). The impact of the delay time was studied in an experimental set acquired with the same approach and retract speed of 10.0 μm/s, and keeping the height constant while in contact, after having reached the sample with a setpoint force *F*_0_ = 1.0 nN. The average detachment work in this condition increases with the delay time over the full range of 120 s (Figure 2c). Conversely, the standard deviation of the data remains almost constant for all points but the last one, which shows a broader distribution of values (Figure 2c). A similar analysis was performed for the setpoint force *F*_0_, while keeping the speeds and the delay time constant, at 10.0 μm/s and 45 s respectively. In this case, only 3 values were acquired, as larger forces were clearly associated with permanent deformations/damage of the cells. The results reported in Figure 2d show that the adhesion work grows with the indentation force, and higher forces are associated with broader distributions.

Furthermore, the impact of two different delay modes, constant-force and constant-height, was also investigated (Figure 3). The cell probe was lowered and retracted at a constant speed of 10.0 μm/s and kept in contact with the target cell for 45 s after the preset force of 1.0 nN was reached. The vertical deflection variation as well as the relative piezo displacement over the time for the two selected delay modes are reported in Figure 3a,b. The results show that, when the setpoint force is kept constant (Figure 3a, blue line) during all the time the cells are in contact, the relative piezo displacement (Figure 3a, green line) decreases due to the dynamic rearrangement of the cells able to remodel their structure when subjected to an external force. On the other hand, when the constant height mode was employed (Figure 3b), after an initial and rapid increase of the vertical deflection to reach the setpoint force of 1.0 nN, the force decreases settling down to lower values (Figure 3b, blue line) being the height to be kept constant as shown by the relative piezo displacement signal (Figure 3b, green line). The distribution of the detachment work while using these two modes is presented in Figure 3c. Results reveal that work needed to detach both cells in the case of constant-force mode is significantly higher than in the case of constant height mode and higher forces are associated with broader distributions.

## 4. Discussion

The pattern of interaction measured with CCFS is extremely rich, and the fine details are expected to depend on the specific cellular system and the characteristics of the molecular players involved in the adhesion [29]. Nevertheless, the general behavior observed in Figure 2 can be described at the first order in terms of simplified components. To the simplest approximation, the CCFS experiment can be modeled as two viscoelastic (semi-)spheres that are brought in contact with constant velocity *v_a_* till the interaction force reaches a maximum value *F*_0_, where the motion is halted for a time τ During the contact, we can assume that a set of *N* surface bonds are formed, and they contribute, together with the intrinsic properties of the cell, to the definition of the detachment work *W* measured while the cells are being pulled apart with constant speed *v_r_*.

We assume that the extraction work *W* is proportional to the number *N* of bonds being formed during the interaction, and that this number in turn simply depends on the area of interaction between the cells and the time they spend in contact:$$W\propto N\propto \int_{T\_i}S\left(t\right)dt$$
where the integral is made over the interaction time *T_i_* and *S*(*t*) indicates the instantaneous interaction surface. Under this simple approximation we can revisit the results presented in the previous section, where only one experimental parameter at a time was modified, keeping all the other conditions constant.

For the experiments of Figure 2c, only the delay time was changed, while approach and retract speed were kept constant. Given that the setpoint force is the same for all these curves, we do not expect a major change of the contact area, and the number N (and so the extraction work W) is expected to be directly proportional to the time spent in contact, plus a small offset associated to the indentation phase (from *F* = 0 to *F* = *F*_0_). The experiments in Figure 2c show this trend, as demonstrated in Figure 4a where the average values of *W* for this experiment, with the corresponding linear fit, is shown.

The same simplified consideration can be applied to guess the trend of *W* as a function of the setpoint force. The maximum force directly affects the area of interaction between the cells. To have an approximation, we can use the Hertz contact mechanics theory [30] that gives an estimate for the area between the surfaces:$$A=\pi a^{2}=\pi R\;\delta_{0}$$
where *a* is the contact radius, *R* the effective radius of the cells, *δ*_0_ the indentation corresponding to the maximum force. The same theory connects this indentation to the force *F*_0_:$$F_{0}\propto \delta^{3/2}\;\to \delta_{0\;}\propto F_{0}{}^{2/3}$$
and this suggests that the area should be proportional to:$$A\propto F_{0}{}^{2/3}$$
and so the adhesion work *W*. In Figure 4b we present the best fit of *W(F*_0_*)* with this power law.

Interestingly, the extraction work appears to be influenced by the approach speed as well (see Figure 2b). In particular, *W* slowly decays with the speed *v_a_*, indicating that the number of established bonds is lower for higher speed. In fact, the time the two cells remain in contact is inversely proportional to the approach speed, as the final (hold) position is the same for all configurations. This behaviour is directly fitted to the data in Figure 4c, where an additional coefficient accounts for constant contributions in addition to the variable one.

To understand the different values of extraction work measured as a function of the retract speed, the number of bonds cannot be the dominating factor, as this is expected to be the same for all conditions. Instead, the origin of the different values has to be found in the dynamical response of the cell–cell complex. While being pulled, this structure will present two main contributions. The first is a proper viscous drag, associated to the nature of the cell, which is expected to be linearly proportional to the pulling speed *v_r_*. While this effect dominates at higher speed, the second mechanism contributes to deviate the curve from the simple linear drag for low speeds. In fact, the two cells are connected by soft bonds, whose resistance is expected to follow a kinetic activation mechanism, which results in a logarithmic dependence on the speed [31]. Putting these two terms together, it is possible to fit the experimental data well with only two parameters (see Figure 4d).

Moreover, the difference between delay modes has been compared and the results are reported in Figure 3. This experiment introduces a different aspect to the analysis, related to the active nature of the adhesion process. Cells are exquisitely mechanosensitive; they are able to feel the external force [32,33] and react by rearranging their shape and reducing the stress. This mechanism cannot be simply described in terms of physical components, but it should include the biological mechanism of mechanosensing, and its main molecular players [34]. Nevertheless, it is crucial to control the contact mode to obtain reliable results in CCFS, as demonstrated in Figure 3.

## 5. Conclusions

This work shows how the choice of the experimental conditions can influence the results derived from CCFS experiments. The specific shape of the CCFS curve depends on a plethora of events associated with the nanoscale interaction between the cells, convolving physical and biological contributions such as the roughness of the membrane, the activation of specific receptors on the surface, or the presence or invaginations and water pockets within the adhesion surface. Although it is not trivial to deconvolve the contribution of any single component contributing to the inherent complexity of the physical contact between the two living cells, we demonstrated that a first-order understanding of the trends observed at different conditions can be deduced by simple theoretical considerations, where the main parameter is the number of bonds being formed between the cells during the contact phase.

Furthermore, this work also suggests that the choice of experimental conditions that are generally not considered, such as the approach speed and the delay mode, can bring significantly, and in some cases unexpected, variations in the results. The hints suggested by our work represent a step forward towards the definition of the best experimental practices.

## Figures and Tables

**Figure 1 sensors-21-01069-f001:**
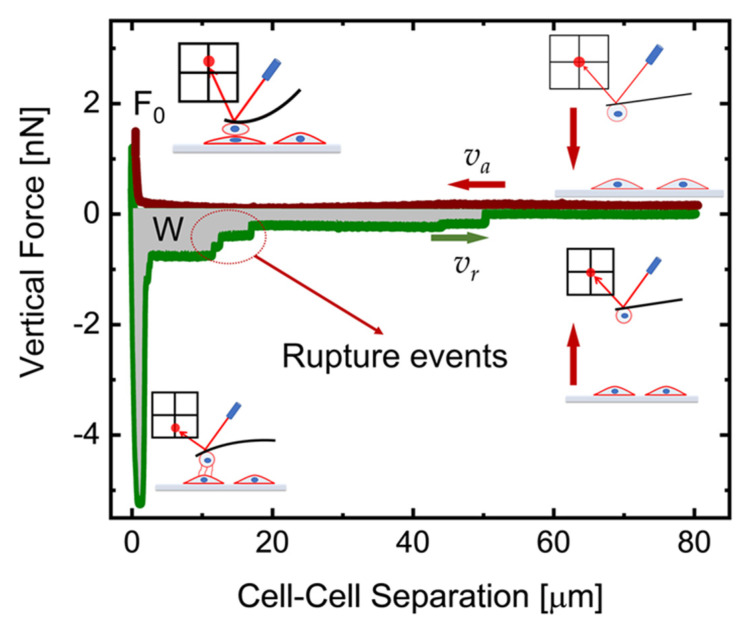
Typical force–distance (F–D) curve acquired in cell–cell force spectroscopy (CCFS). The curve starts when the CHO cell attached to a tipless cantilever is positioned at 80 μm on top of a selected target CHO cell. This is represented as the cartoon in the top right. The atomic force microscope (AFM) cantilever then begins to move towards the target cell with a constant approach speed *v_a_* (red line) reaching the preset setpoint force value *F*_0_ (see cartoon in the top left). At this point, cells are kept in contact having constant either the applied force or the position (delay mode). After the contact time (delay time, τ), the AFM cantilever is pushed-away from the target cell and cells begin their detachment (cartoon in the lower left). The detachment process (green line) occurs with a constant retract speed *v_r_* and it is characterized by several rupture events. The latter are generally associated with specific cell-cell interactions. The F–D curve ends when a final plateau at force ~0 nN is reached corresponding with the full detachment of the cells.

**Figure 2 sensors-21-01069-f002:**
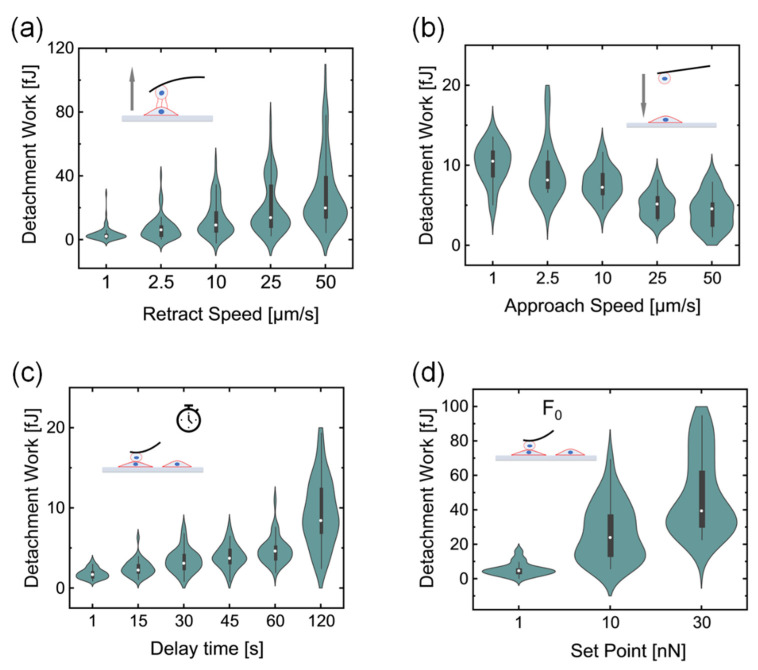
Influence of the experimental settings on cell-cell interactions. (**a**,**b**) show the distribution of the work spent to detach the probe cell from the target cell when the retraction speed (**a**) and the approach speed (**b**) were modified. (**c**,**d**) display the effect of the delay time τ (**c**) and setpoint force (**d**) in the distribution of the adhesion work.

**Figure 3 sensors-21-01069-f003:**
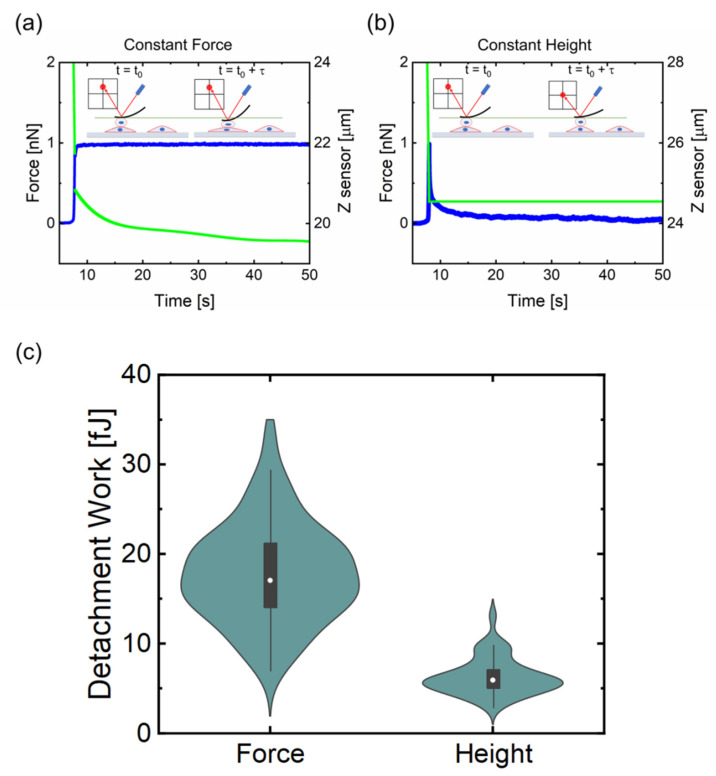
Influence of the two different delay modes on cell–cell interactions. When cells are kept under a constant applied force during the contact time (blue line in (**a**)), they reshape their structure resulting in a compensatory decrease in the relative piezo displacement signal (green line). In contrast, when it is kept constant the separation between the AFM cantilever and the sample (green line in (**b**)), the vertical deflection signal in the photodiode initially increases to reach the preset force setpoint (blue line) to, subsequently, decrease during the contact time. (**c**) shows the distribution of the work spent to detach the probe cell from the target cell when both delay modes are investigated.

**Figure 4 sensors-21-01069-f004:**
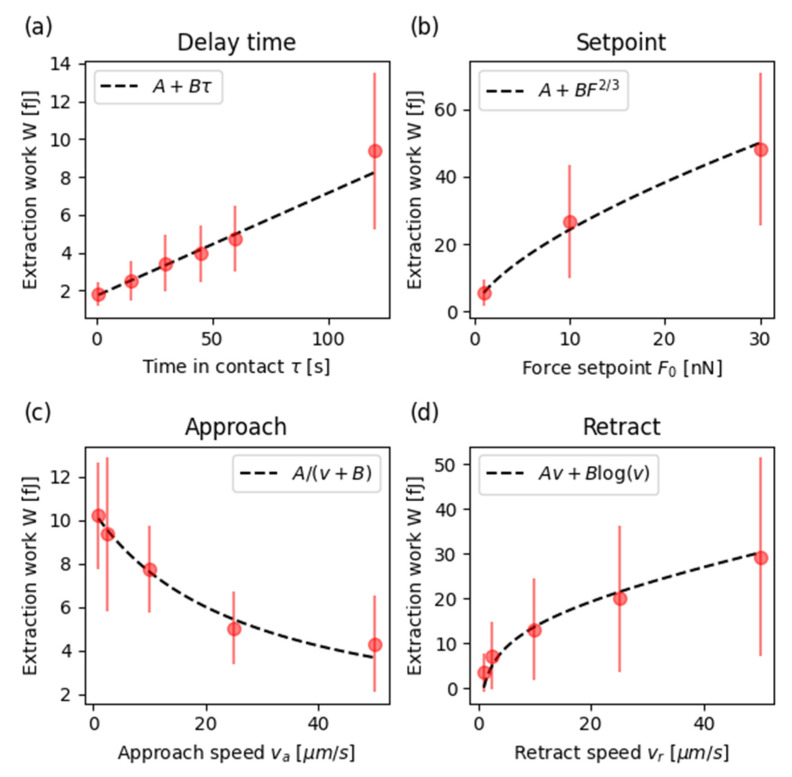
Average extraction work *W* as a function of (**a**) delay time, (**b**) force setpoint, (**c**) approach speed and (**d**) retract speed. The experimental data were fitted with the equation reported in the corresponding legend, obtaining the following parameters: (**a**) A = 0.05 fJ and a slope B = 1.72 fJ/s; (**b**) offset A = 0.42 fJ and a coefficient B = 5.13 fJ N^3/2^; (**c**) A = 282 zJ/s and B = 27.0 μm/s; (**d**) A = 0.21 nPa s and B = 5.0 fJ.

## Data Availability

The data presented in this study are openly available in the Enlighten: Research Data repository of the University of Glasgow at doi:10.5525/gla.researchdata.1103.

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
