# Peer review of "Impact of Experimental Parameters on Cell–Cell Force Spectroscopy Signature"

_sensors, 2021, doi:10.3390/s21041069_

Round 1

Reviewer 1 Report

This is an impressive presentation of how various parameters impact cell-cell-force spectroscopy (CCFS) signatures. But his approach doesn’t take into account cell-cell receptor interaction. The main drawback of CCFS is that one cannot segregate specific signatures between two cells that are responsible for a particular physiological response. Though this study reports the impact of various parameters that impact net adhesion between cells when they are subjected forced physical contact.

 Following clarifications are required –

  • What is the N for each experiment ? e.g. for 30 nN Set Point measurements how many measurements were done.  How many force curves were collected for each experiment?
  • Page 8 – “In fact, in this case the maximum force was kept constant, and so the surface area is approximately the same for all CCFS”. This statement holds true for non-biological/non-cellular interactions but not at the interface of two cells. The cell on the surface has a very flat morphology and the cell attached to the cantilever will not have perfect semi-sphere morphology. Plus depending on the approach speed, delay time, and set point surface area will greatly differ. This is the very reason you are observing differences in the  work spent when the surface of the cells detach from each other. As stated in your article “Work W is proportional to the number N of bonds being formed during the interaction”. Here the number of bonds formed during interaction will change according to the area of contact between the surfaces. Plus there can be a number of water pockets between the two cell membranes depending on how fast and with what force two cell membranes come in contact with each other. Cells have a very dynamic cytoskeletal network and this can be observed in constant height and constant force data in figure 3(c). Therefore, it’s difficult to agree with this statement that the “ surface area is same for all CCFS”, even for the constant applied force measurements. This statement will require fluorescence data that can map surface area of contact at the interface of two cell membranes.
  • Double full stops at the end of the abstract.
  • Page 3 - "density of 3 × 103cells ml−1” , 10 raised to power 3 needs to be corrected or its just 300 cells per ml.
  • Reference 25 is incomplete.

Author Response

Response to specific comments

We thank the reviewers for the careful reading and for their suggestions. We attach here a revised version of the manuscript, with the main changes reported in red. The specific replies to the comments raised by the reviewers are reported below. Reviewer’s comments are reported in blue, with the corresponding reply in black.

Reviewer 1

  1. What is the N for each experiment? e.g. for 30 nN Set Point measurements how many measurements were done. How many force curves were collected for each experiment?

This has been fixed, indicating N for all experiments in the corresponding experimental sections.

  1. Page 8 – “In fact, in this case the maximum force was kept constant, and so the surface area is approximately the same for all CCFS”. This statement holds true for non-biological/non-cellular interactions but not at the interface of two cells. The cell on the surface has a very flat morphology and the cell attached to the cantilever will not have perfect semi-sphere morphology. Plus depending on the approach speed, delay time, and set point surface area will greatly differ. This is the very reason you are observing differences in the  work spent when the surface of the cells detach from each other. As stated in your article “Work W is proportional to the number N of bonds being formed during the interaction”. Here the number of bonds formed during interaction will change according to the area of contact between the surfaces. Plus there can be a number of water pockets between the two cell membranes depending on how fast and with what force two cell membranes come in contact with each other. Cells have a very dynamic cytoskeletal network and this can be observed in constant height and constant force data in figure 3(c). Therefore, it’s difficult to agree with this statement that the “ surface area is same for all CCFS”, even for the constant applied force measurements. This statement will require fluorescence data that can map surface area of contact at the interface of two cell membranes.

We thank the reviewer for the comment, and we appreciate this sentence was not put in the correct context. The main point is that the extraction work W is mainly associated to the number N of bonds created during the interaction, and this number N depends on many parameters - as highlighted by the reviewer. To isolate the contribution of each parameter, we performed experimental sets in which only one of the main effectors was modified (say, in this case, only the force was changed, not the speed). This aspect has been clarified in the new text (see below). Moreover, the reviewer correctly points out the fact that many phenomena can contribute, such as the roughness of the membrane and the formation of pockets, and this aspect has been added in the discussion section as well. Nevertheless, our aim was to identify the leading contribution to W, and we think that the argument about W being proportional to N is a useful one to provide a first insight in the experimental results (and a guide to select the experimental parameters).

New sentence in the paragraph highlighted by the reviewer:

Under this simple approximation we can revisit the results presented in the previous section, where only one experimental parameter at a time was modified, keeping all the other conditions constant.
For the experiments of Fig. 2c, only the delay time was changed, while approach and retract speed were kept constant. Given that the setpoint force is the same for all these curves, we do not expect a major change of the contact area, and the number N (and so the extraction work W) is expected to be directly proportional to the time spent in contact, plus a small offset associated to the indentation phase

New sentence in the discussion: 

The specific shape of the CCFS curve depends on a plethora of events associated to the nanoscale interaction between the cells, convolving physical and biological contributions such as the roughness of the membrane, the activation of specific receptors on the surface, or the presence or invaginations and water pockets within the adhesion surface. A We demonstrated that, although it is not trivial to deconvolve the contribution of any single component contributing to the the causes of these variations are hidden by the inherent complexity of the physical contact between the two living cells, we demonstrated that a good first order understanding of the trends observed at different conditions can be deduced by simple theoretical considerations, where the main parameter is the number of bonds being formed between the cells during the contact phase.

  1. Double full stops at the end of the abstract.
  2. Page 3 - "density of 3 × 103cells ml−1” , 10 raised to power 3 needs to be corrected or its just 300 cells per ml.
  3. Reference 25 is incomplete.

All these typos and inconsistencies were fixed in the revised version. Thanks for the careful reading and sorry for this.

Reviewer 2 Report

In this manuscript the authors describe cell-cell force spectroscopy methodology and how different experimental parameters impact the experimental measurements. Through systematic examination the authors clearly show how different approach speeds, delay time, feedback mode and force change the detachment work. I enjoyed reading this paper, and while this is not the first study of this kind, I feel that this work is very beneficial to the field. The experiments are rigorous and supported by the statistics. I have a few minor comments.

I think that the term set point might not be clear to non-experts in AFM and could be defined early in the text and methods.

In figure 4b where the authors plot the scaling of extraction work vs setpoint, the three points in this graph could likely be well fit with a range of different models given the error bars. The authors could report a metric for goodness of fit for this model.

Methods 2.2 introduces the procedure for attaching the cell to the cantilever and references an agarose spot? Can this be described in a little more detail. The text also reads 3x103 cells in a number of places, I assume the final 3 should be superscript.

2.4 Data Analysis where the Savitzky-Golay filter is introduced, there is a number in square brackets?

Figure 1 caption "and it" is in italics

Author Response

Response to specific comments

We thank the reviewers for the careful reading and for their suggestions. We attach here a revised version of the manuscript, with the main changes reported in red. The specific replies to the comments raised by the reviewers are reported below. Reviewer’s comments are reported in blue, with the corresponding reply in black.

Reviewer 2

  1. I think that the term set point might not be clear to non-experts in AFM and could be defined early in the text and methods.

We agree with the reviewer, and we extended the text on page 4 to clarify what we mean with setpoint force.

  1. In figure 4b where the authors plot the scaling of extraction work vs setpoint, the three points in this graph could likely be well fit with a range of different models given the error bars. The authors could report a metric for goodness of fit for this model.

We agree with the reviewer that 3 points are not enough to demonstrate any non-linear scaling. Our aim here is more generically to demonstrate that the simple assumption that W depends on N catches the main trend of the experiments in all conditions. Nevertheless, we expect that a finer analysis will highlight higher order contributions and the effect of local heterogeneity. This concept has been clarified in the discussion section:

  1. Methods 2.2 introduces the procedure for attaching the cell to the cantilever and references an agarose spot? Can this be described in a little more detail.

We thank the reviewer that highlighted this important point. Actually, the use of the agarose to increase the efficiency of the cell capture procedure is strongly recommended since it facilitate one of the fundamental step in SCFS experiments. We extended this point at page 3:

Before cell seeding, a small part of the coverslip was coated with agarose, by spreading a 20 µL drop of 0.15% w/w agarose solution (Sigma-Aldrich) until gelification. The lack of adhesion between the cell and the repulsive agarose surface significantly increased the efficiency of the cell capture procedure.

  1. The text also reads 3x103 cells in a number of places, I assume the final 3 should be superscript.
  2. Data Analysis where the Savitzky-Golay filter is introduced, there is a number in square brackets?
  3. Figure 1 caption "and it" is in italics

All these typos and inconsistencies were fixed in the revised version